# Effectiveness of interventions for people living with dementia and their carers in Chinese communities: protocol for a systematic review and meta-analysis of randomised controlled trials

Cheng Shi [1,2] Shuangzhou Chen,[1] Maximilian Salcher-Konrad [3]
Jacky C P Choy,[1] Hao Luo [1] Dara Kiu Yi Leung [1] Xinxin Cai,[1] Yue Zeng,[1]
Ruizhi Dai,[4] Adelina Comas-Herrera,[3] David McDaid,[3] Martin Knapp,[3]
Gloria Wong [1]

For numbered affiliations see end of article.

**Correspondence to**
Dr Gloria Wong;
ghywong@hku.hk

## ABSTRACT

**Introduction** As the largest and most rapidly ageing population, Chinese people are now the major driver of the continued growth in dementia prevalence globally. The need for evidence-based interventions in Chinese communities is urgent. Although a wide range of pharmacological and non-pharmacological interventions for dementia have been trialled in Chinese populations, the evidence has not been systematically synthesised. This systematic review and meta-analysis aims to map out the interventions for people living with dementia and their carers in Chinese communities worldwide and compare the effectiveness of these interventions.

**Methods and analysis** This protocol followed the Preferred Reporting Items for Systematic Review and Meta-Analysis Protocols checklist. We will search Chinese (China National Knowledge Infrastructure, WanFang DATA) and English bibliographical databases (MEDLINE, EMBASE, PsycINFO, CINAHL Plus, Global Health, WHO Global Index Medicus, Virtual Health Library, Cochrane CENTRAL, Social Care Online, BASE, MODelling Outcome and cost impacts of interventions for DEMentia (MODEM) Toolkit, Cochrane Database of Systematic Reviews), complemented by hand searching of reference lists. We will include studies evaluating the effectiveness of interventions for dementia or mild cognitive impairment in Chinese populations, using a randomised controlled trial design, and published between January 2008 and June 2020. We will use a standardised form to extract data and Version 2 of the Cochrane risk-of-bias tool for randomised trials to assess the risk of bias of the included studies. Collected data will be fully interpreted with narrative synthesis and analysed using pairwise and network meta-analyses to pool intervention effects where sufficient information is available. We will perform subgroup analysis and meta-regression to explore potential reasons for heterogeneity.

**Ethics and dissemination** No formal ethics approval is required for this protocol. The findings will facilitate the development of studies on interventions for dementia and timely inform dementia policymaking and practice. Planned

### Strengths and limitations of this study

► This systematic review and meta-analysis will be the first review of randomised controlled trials (RCTs) on the effectiveness of both pharmacological and non-pharmacological interventions for people living with dementia and their carers in Chinese communities worldwide.
► We will use a comprehensive search strategy of publications in both Chinese bibliographical databases (China National Knowledge Infrastructure, WanFang DATA) and English bibliographical databases (MEDLINE, EMBASE, PsycINFO, CINAHL Plus, Global Health, WHO Global Index Medicus, Virtual Health Library, Cochrane CENTRAL, Social Care Online, BASE, MODelling Outcome and cost impacts of interventions for DEMentia (MODEM) Toolkit, Cochrane Database of Systematic Reviews).
► We will narratively synthesise the collected data to map out the dementia-related interventions studied in Chinese communities and conduct pairwise and network meta-analyses to compare the effectiveness of interventions.
► This review will be limited by the number and quality of RCTs conducted in Chinese communities.

dissemination channels include peer-reviewed publications, conference presentations, public events and websites.
**PROSPERO registration number** CRD42019134135.

## INTRODUCTION

Around 50 million people currently live with dementia worldwide, of whom 20% are Chinese populations.[1] Chinese population refers to people of Chinese ethnicity or national heritage, regardless of their nationality or region of residence. As the largest and most rapidly ageing population, the Chinese

are now the major driver in the continued growth of global dementia prevalence.[2] Due to the physical and emotional challenges involved in caring, dementia affects not only people living with the condition but also their families, formal carers and other supporters.[3] With a culture emphasising filial piety, coupled with insufficient care services, family care is often the main supporting resource for people living with dementia (PLwD) in Chinese communities worldwide. Dementia has been recognised as one of the most burdensome diseases among Chinese populations.[4]

There is currently no cure for dementia, although symptoms can be managed with effective intervention and good care.[3] China recently launched its national dementia strategy, one of whose main tasks is to improve the well-being of PLwD by increasing service provision.[5] Taiwan updated its dementia policy in 2017, promoting dementia research, innovation and development as one of its seven strategies.[6] In Macau's 10-year Plan of Action on Dementia Services published in 2016, strengthening community services and caregiver support comprises one of its five strategies.[7] In Hong Kong, a government service review and programme plan published in 2017[8 9] highlighted the need to strengthen services for PLwD and recommended a seven-stage model for dementia service following the WHO and Alzheimer's Disease International's framework.[10] The need for evidence-based interventions and care services in Chinese populations is urgent.

Studies on dementia interventions appear to be scarce in Asian populations.[2] Most evidence on drug treatment and non-pharmacological interventions has been generated in Western countries, with questionable relevance for Chinese populations. For example, cognitive stimulation therapy (CST) used alone or in combination with medication was shown to be effective and even cost-effective in improving cognition and quality of life,[11–14] leading to a recommendation for routine use by England's National Institute for Health and Care Excellence[15] and by Alzheimer's Disease International.[16] In contrast, preliminary findings from a study applying CST with Hong Kong Chinese suggest that a larger number of participants needed to be treated to achieve clinically significant improvement in cognition.[17] Such discrepancies in an intervention's effect, possibly due to cultural differences, highlight the importance of generating evidence on the effectiveness of dementia-related interventions relevant to local populations.

There is now increasing evidence on a wide range of interventions for dementia undertaken in Chinese populations. A few reviews have been published, focusing on specific interventions and subtypes of dementia, such as the efficacy of donepezil in Chinese with Alzheimer's disease,[18] Chinese herbal medicine as adjunctive therapy for vascular dementia[19] and traditional Chinese mind-body exercise (baduanjin) in older adults with mild cognitive impairment (MCI).[20] Growing evidence also suggests that the therapeutic response to dementia intervention (eg, donepezil) might differ between Chinese and Western populations due to pharmacogenetic factors,[18 21] thus emphasising the need for more accurate evaluations of interventions tailored to Chinese populations.[22]

Some existing and ongoing studies aim to synthesise evidence for dementia intervention and care, including the Modelling Outcome and Cost Impacts of Interventions for Dementia (MODEM) project[23] with a dementia evidence toolkit (https://www.modem-dementia.org.uk/) covering dementia interventions in English literature and the Strengthening Responses to Dementia in Developing Countries (STRiDE) project (https://stride-dementia.org/) with an ongoing systematic review and meta-analysis on the evidence in seven low- and middle-income countries.[24] There is no comprehensive evidence synthesis on the effectiveness of dementia or dementia-related interventions that cover different types of dementia (eg, Alzheimer's disease, vascular dementia, frontotemporal dementia, Lewy body disease and mixed dementia) and interventions (eg, pharmacological treatment, psychosocial intervention and traditional Chinese medicine) conducted in Chinese populations. Existing systematic reviews have focused mainly on the English literature, where evidence from high-income areas such as Hong Kong and Taiwan can be found. Although Chinese academic databases have been recognised as a valuable resource for dementia-related studies, they have not been fully explored.[25–27]

To our knowledge, this will be the first systematic review and meta-analysis to comprehensively synthesise and assess the evidence on the effectiveness of interventions for PLwD and their carers among Chinese populations in Chinese and English bibliographical databases. We aim to (1) map out interventions for dementia studied in Chinese communities, and (2) compare the effectiveness of those interventions for achieving desired outcomes. This study will contribute to shape the understanding of existing evidence on effectiveness of dementia-related interventions, improve quality of life of PLwD and their carers and provide valuable information for practice, policymaking and further research. As part of a research project, Tools to Inform Policy: Chinese Communities Actions in Response to Dementia (TIP-CARD; www.tip-card.hku.hk/), this study also aligns with the above-mentioned dementia evidence synthesis effort by the STRiDE project.[24]

## METHODS AND ANALYSIS
### Protocol and registration
This protocol for this systematic review and meta-analysis followed the Preferred Reporting Items for Systematic Review and Meta-Analysis Protocols checklist.[28] This study has been registered on the PROSPERO platform (www.crd.york.ac.uk/prospero).

### Eligibility criteria
#### Population
We will include studies conducted among adults (aged 18 years and over) living with dementia or MCI and their

carers in Chinese populations. We will include relevant studies conducted in any type of care settings, such as home, community, residential homes, clinics, hospitals and other care settings. Participant characteristics such as gender, education and age at diagnosis will not be used for excluding studies.

We will include studies covering people living with any type and stage of dementia. Dementia, as a major neurocognitive disorder, describes a group of symptoms of cognitive decline, including, but not limited to, Alzheimer's disease, vascular dementia, frontotemporal dementia, Lewy body disease and mixed dementia. Studies conducted among people living with MCI, mild neurocognitive disorder, vascular cognitive impairment and no dementia will be eligible for inclusion due to the higher risk of developing dementia in later years.[29] We will also include studies conducted among people with diseases cooccurring with dementia or MCI, and people with dementia or MCI with unknown subtype, as long as the diagnostic criteria for dementia or MCI were explicated.

Our definition of dementia carer refers to persons involved in care provision and management and will not depend on whether or not the carer is paid, lives with the person they care for or provide direct or indirect care. Therefore, dementia carers include health and social care professionals, care managers, care workers, administrative staff of care facilities, family carers, other unpaid carers and family members assisting with care decisions. We will focus on studies conducted among people of Chinese ethnicity or national heritage regardless of their nationality or location of residence. Studies without explicating the proportion of Chinese participants over 50% or studies without a specific subgroup analysis for Chinese participants will be excluded.

### Intervention
Based on the effectiveness perspective,[24 30] any type of interventions for improving desired outcomes will be eligible. We will include studies on pharmacological treatment, non-pharmacological intervention (eg, cognitive intervention, technological intervention, training and exercise) or multicomponent interventions. We will exclude studies: (1) where no clear intervention was described, (2) on primary prevention of dementia and (3) on non-interventional studies.

### Comparison
Given the broad range for interventions of interest, any comparisons within the context of eligible study design will be acceptable for inclusion, such as active comparators, treatment as usual, placebo and no treatment.

### Outcomes
Any type of outcomes of dementia-related intervention will be eligible for inclusion from the perspective of effectiveness, which may affect individuals, families, the dementia care workforce, wider society and social or healthcare systems. Dementia often triggers complex problems in many domains.[22] According to the MODEM dementia evidence toolkit (https://www.modem-dementia.org.uk/), outcomes measured in existing studies may include (1) cognition, behavioural and psychological symptoms, functional status, physical health and quality of life of PLwD, (2) carer burden, carer's mental health, quality of life and other carer outcomes (eg, financial burdens), (3) service use, cost reduction (including hospital use reduction and care home admission delay) and service satisfaction, (4) risk reduction (of dementia and comorbidities) and prevention or management of comorbidities. To capture the diversity of interventions trialled in Chinese communities, we will accept all outcome measures that reflect intervention effectiveness.

### Study design
To identify potential causal relationships, we will only include studies using randomised controlled trial (RCT) or cluster RCT designs. To control study quality, we will only include RCTs with a low risk of bias (RoB) in the process of evidence generation. According to Version 2 of the Cochrane RoB tool for randomised trials,[31] methods, used for generating random allocation sequence indicating low RoB, include computer-generated random numbers, a random number table, coin tossing, shuffling cards or envelopes, throwing dice or drawing lots. Studies that use no random element or provide no information on the generation process of the random allocation sequence will be excluded.

To minimise small-study effects,[32 33] we will exclude studies with a sample size of less than 50 in either the intervention group or comparison group(s) for the eligible population. For studies conducted with a population of mixed ethnicity, the sample size of each study arm for Chinese subgroup analysis should be greater than 50 participants. For studies in which more than 50% of participants are Chinese and all participants are randomly grouped, the sample size of each study arm is expected to be greater than 50 participants regardless of ethnicity.

### Publication type
We will include the primary publications of intervention studies and grey literature evaluating the effectiveness of dementia-related interventions in Chinese populations. Relevant systematic reviews or scoping reviews will be included in the first step of screening and then will be used to complement the primary publications by hand searching of reference lists. Conference abstracts will be included if they contain sufficient information to assess eligibility for inclusion.

### Publication period
Studies published between January 2008 and June 2020.

### Language
Studies will be limited to English and Chinese publications.

## Information sources

We will search two major Chinese bibliographical databases (China National Knowledge Infrastructure and WanFang DATA) and English bibliographical databases (MEDLINE, EMBASE, PsycINFO, CINAHL Plus, Global Health, WHO Global Index Medicus, Virtual Health Library, Cochrane CENTRAL, Social Care Online, BASE, MODEM Toolkit, Cochrane Database of Systematic Reviews). Hand searching of reference lists among review studies will complement the database searches.

## Search strategy

We will adapt an established search strategy protocol[24] used to search for English language literature. Corresponding Chinese search terms have been translated and adapted by three bilingual researchers (GW, SC and CS) experienced in dementia/ageing research with a training background in psychology, psychiatry, translation, social work and social policy from Hong Kong and mainland China. Search terms in English and Chinese are listed in table 1. In studies published in English, the search terms related to Chinese populations include 'China', 'Chinese', 'Sino', 'Hong Kong', 'Taiwan', 'Taiwanese', 'Macau' and 'Asian'.

For studies published in English, we will first extract eligible study records identified from an ongoing systematic review,[24] which used the same search strategy and search terms for dementia intervention and covered studies published between 2008 and 2018. Then, we will search these terms for Chinese populations in the title, abstract and keywords. Second, we will repeat the English bibliographical database search mentioned above to identify studies published between January 2019 and June 2020.

For studies published in Chinese, we will use Python,[34] a programming language, to facilitate Chinese bibliographical database searching by using dementia-related search terms (search items number 1–4 in table 1). This is because of the technical challenge posed by limitations on the number of search terms and exported records per time in the two Chinese bibliographical databases. The search results for dementia-related study records will be exported in a Microsoft Excel spreadsheet. Then, we will search the intervention-related terms (search items number 5–53) in the title and abstract.

## Study records

### Data management

To deal with a potentially large number of search results and various data sources in two languages, we will manage references using two web-based software packages during the review and extraction process: (1) Rayyan (https://rayyan.qcri.org/), a web and mobile app that can facilitate the initial screening of abstracts and titles using a semiautomated process[35] and (2) Covidence (https://www.covidence.org/), an internet-based software platform for managing systematic reviews, including study selection, RoB assessment and data extraction.

Duplicate publications will be checked based on title, author, journal and year using Microsoft Excel, and the 'Find duplicates' function in Rayyan and Covidence. Multiple publications from the same study will be identified based on the key information (eg, authors' names, study design, intervention and outcomes) from the full texts or by contacting authors for clarification if needed. Once confirmed, included multiple publications will be linked on Covidence.

## Study selection

Study selection will be a two-step process, with detailed explanations for inclusion and exclusion criteria in each step. First, two researchers will independently screen the title and abstract and determine the study's inclusion or exclusion on Rayyan. A justification (criterion) will be required for any exclusion decision. Studies with insufficient information in the title and abstract to enable a decision to be made will be included at this stage. The Rayyan machine learning-based classifier[35] will be considered to facilitate the title and abstract screening, given the potentially work overload. Using a certain number of manually screened studies as a training data set, Rayyan will generate a relevance rating for each study, ranging from 0.5 (lowest) to 5 (highest).[35] We may use a low relevance score (eg, below 1.5) as a threshold to guide study exclusion.

Second, studies included after title and abstract screening will be uploaded to Covidence for full-text review by two independent reviewers, who will provide a justification for each excluded study. Review studies will be excluded at this stage, although their reference lists will be used to complement the database search results.

All disagreements in each step will be resolved through discussion between the two reviewers. If consensus is unreachable, a third reviewer will be consulted for a final decision.

Reviewers for title and abstract screening and full-text review will be able to read and understand inclusion/exclusion criteria for publications in both English and simplified Chinese.

On completion of the selection process, we will generate a Preferred Reporting Items for Systematic Reviews and Meta-Analyses flowchart[36] to illustrate the inclusion and exclusion of studies at each stage in study selection.

## Data collection process

We will use a standardised form based on the template available from Covidence for data extraction that will be pilot tested using included studies. To ensure data consistency across reviewers, we will organise exercises and group discussions for reviewer training. Due to the anticipated large number of potentially eligible studies, the data extraction form will be completed by one reviewer and verified by the second reviewer. We will keep all records of corrections or amendments to the data extraction. For studies that do not report the required information, we will contact the authors to request information.

**Table 1** Search terms related to dementia and intervention in English and simplified Chinese

| Search number | Search terms in English | Search terms in simplified Chinese |
|---|---|---|
| 1 | Dementia | 痴呆 or 失智 or 认知症 |
| 2 | Cognitive disorder | 认知障碍or认知功能障碍or认知紊乱or认知功能紊乱 |
| 3 | Alzheimer | 茨海默 or 兹海默 |
| 4 | ((cognit* or memory or cerebr*) adj3 (impair* or los* or declin* or deteriorat* or degenerat*)).mp. | (认知or记忆or脑) (缺损or缺失or退*化or衰退or 下降 or 损伤 or 恶化 or 损害 or 退行) |
| 5 | (Intervention* or therap* or treatment* or program* or manage* or prevent* or diagnos* or polic*).mp. | 干预 or 介入 or 治疗 or 疗法 or 方案or 处理 or 预防 or 诊断 or 措施 or 手段 or 政策or应用 or支持or效果or疗效or观察or价值or临床or分析 |
| 6 | Cognitive therapy | 认知 (治疗or疗法) |
| 7 | Cognitive stimulation | 认知 (刺激or促进) |
| 8 | Cognitive training | 认知训练 |
| 9 | Cognitive rehabilitation | 认知 (复康or复健or康复) |
| 10 | Drug therapy or pharmacotherapy | *药* |
| 11 | Cholinesterase inhibitors | 胆碱分解抑制剂 or 胆碱酵素抑制剂 or 胆碱酶抑制剂 |
| 12 | Cholinesterase agent | 胆碱分解剂 or 胆碱酵素剂 or 胆碱酶剂 |
| 13 | (Sedative or tranquili* adj3 (agent* or drug*)).mp. | (镇静 or 镇定or 安神 or 安定) (药 or 剂) |
| 14 | Antipsychotic or neuroleptic (agent* or drug*) | 抗精神病 (药 or 剂) |
| 15 | exp Serotonin Reuptake Inhibitors or ssri | (血清素 or 5-羟色胺) (再摄取 or 再吸收 or回收) 抑制剂 |
| 16 | Benzodiazepines | 苯二氮平 or 苯二氮卓 |
| 17 | (memantine or donepezil or rivastigmine or galantamine or souvenaid or risperidone or haloperidol or olanzapine or quetiapine or citalopram or dextromethorphan or carbamazepine or mirtazapine or sertraline or moclobemide or trazodone or melatonin or ramelteon or methylphenidate).mp. | (美金刚 or 美金胺) (多奈哌齐 or 多奈呱其) (卡巴拉汀or利斯的明) (加兰他敏or加兰他明or格兰他明) (智敏捷) (利培酮or 利螺环酮) (氟哌啶醇or氟哌丁苯or氟哌醇or卤吡醇) (奥氮平) (喹硫平) (西酞普兰) (右美沙芬or右旋美沙酚or右旋美索芬or右甲吗喃) (卡馬西平or 卡马平 or 卡巴氮平 or 卡巴马平) (米氮平) (舍曲林) (吗氯贝胺) (曲唑酮) (褪黑素 or褪黑激素) (雷美替胺 or拉米替隆) (哌甲酯 or 派醋甲酯 or盐酸甲酯) |
| 18 | Movement Therapy | (运动or 动作) |
| 19 | (Physical activit* or physical training).mp. | (运动or体育 or 体能) (活动 or 训练) |
| 20 | (social adj3 activit*).mp. | 社交活动 or 社会活动 |
| 21 | Psychotherapy | 心理 (治疗or疗法) |
| 22 | (behavio?r* adj3 therap*).mp. | 行为 (治疗or疗法) |
| 23 | Counseling | 辅导 or 咨询 |
| 24 | ((Psychosocial or psycho social) adj3 (support or interven* or care)).ti,ab. | (社会心理or社交心理） (支援or治疗or干预or介入or照顾) |
| 25 | Alternative medicine | (替代or另类) (治疗or疗法or医学or医疗) |
| 26 | Chinese medicine | 中医 or 中药 |
| 27 | Acupuncture | 针灸or针刺or电针 |
| 28 | (herb* adj3 (tea or remedy or remedies or medicine*)).ti,ab. | 草药or 药草 or药用植物or 草本 or 茶疗 |
| 29 | Gingko | 银杏 or 白果 |
| 30 | homeopathy | (顺势 or 同质 or 同种) |
| 31 | ((music or art or aroma or light or photo or pet or pets) adj3 therap*).ti,ab. | (音乐 or 艺术 or 香薰 or 光照or光线or宠物or动物or 舞蹈) |
| 32 | Massage | 按摩or推拿 |
| 33 | Mind Body Therapy | 身心or 心身 or 正念or冥想 |
| 34 | Advance directives | 预设医疗指示 or 预设指示 or 预前意愿 or 预先指示 |
| 35 | (Advance? adj3 (care or medical or healthcare) adj3 plan*).mp. | (预设 or 预立) (护理计划 or 临终*计划 or 医疗决定) |
| 36 | (decision* adj3 (aid* or support)).mp. | 决策援助 or 决策辅助 or 决策支持 |
| 37 | Case Management | 个案管理 |
| 38 | (communicati* adj3 skill* adj3 training).mp. | 沟通技巧 (训练 or 培训) |
| 39 | (dementia care adj3 map*).mp. | 认知障碍症照顾图谱 or 老年痴呆症照顾图谱 or 失智症照顾测绘 |
| 40 | ((person* or patient*) adj3 cent* adj3 care).mp. | (以人为本or 人本 or 以人为中心 or 病人为本) (照顾 or 照护 or 护理 or 治疗 or 医疗) |
| 41 | ((caregiver or carer) adj3 educat*).mp. | (照顾者or 家属or 家庭or照护者or 照料者) 教育 |
| 42 | Support Groups | (支援or支持or互助） (小组 or组 ) |
| 43 | Self-Help Techniques | 自助法 or 自助*法 or 自助技巧 or 自救* |

Continued

**Table 1** Continued

| Search number | Search terms in English | Search terms in simplified Chinese |
| --- | --- | --- |
| 44 | Social Support | 社交支援/社交支持/互助组 |
| 45 | Computer assisted diagnosis | 电脑辅助诊断/计算机辅助诊断 |
| 46 | Telemedicine | 远程医疗 or远距医疗 or 远距离医疗 |
| 47 | Computer Assisted Therapy | 电脑辅助治疗or 计算机辅助治疗 |
| 48 | Mobile Devices | 移动设备 or 行动装置 |
| 49 | ((smart adj2 (phone* or device* or tablet*)) or smartphone*).mp. | 智能 (手机or电话or平板电脑) or 智能* or 可穿戴* |
| 50 | cognitive aid | 认知辅助/认知帮助 |
| 51 | Reminder | 提示 or 提醒 |
| 52 | Robot | 机器人 |
| 53 | Animal Experiment | 动物实验/动物试验 |

We will prepare the data extraction form in English. For studies in Chinese, our bilingual reviewers will complete data extraction using the original expression in Chinese full texts except for the outcome name and brief introduction of the intervention, which will be recorded in English based on the English abstract if available or manual translation. The final extracted evidence from both the English and Chinese studies will be verified by one bilingual researcher (CS) to ensure consistent translation.

### Data items

We will extract information on items listed in box 1 from the included studies.

### Outcomes and prioritisation

In line with our research aims, we will first record all types of outcome and outcome measures stated in the included studies to map out the dementia-related interventions conducted in Chinese communities. Due to the anticipated number of Chinese studies from an ongoing review,[24] we will prioritise the following outcomes of interest when extracting outcome results from included studies.

As dementia is a condition affecting cognition by definition, we will prioritise outcome on changes in cognition. Common assessments for measuring cognitive impairment level or performance include the Mini–Mental State Examination (MMSE), the Montreal Cognitive Assessment (MoCA) and the Alzheimer's Disease Assessment Scale-Cognitive Subscale (ADAS-Cog). Given the effects of dementia on the ability to organise activities,[22] we will also focus on changes in functional ability following treatment. For example, the Disability Assessment for Dementia is designed for evaluating functional ability to complete activities of daily living (ADL) and instrumental ADL among PLwD.

Caring for a person living with dementia can be very stressful, which may lead to a higher level of depression or health issues.[37] For studies conducted among carers of PLwD, we will focus on changes in quality of life and carer burden,[38 39] as measured by tools such as the

---

**Box 1   Data to be extracted from included studies**

General information
▸ Reviewers' name.
▸ Date of data extraction.
▸ Publication details and identification.
▸ Sponsorship source.
▸ Research site: places (city-level) where the trial was conducted.
▸ Setting (eg, hospital, care home, community).
▸ Study aim(s).
▸ Publication language: Chinese or English.

Methods
▸ Study design.
▸ List of all outcomes with instruments reported in the study.

Population
▸ Inclusion criteria.
▸ Exclusion criteria.
▸ Group differences.
▸ Clinical features (eg, types of dementia, severity and duration of dementia).
▸ Baseline characteristics of participants in each study arm or overall participants: demographics (eg, age, gender), socioeconomic status (eg, education), clinical outcomes if any, number of participants.

Intervention
▸ Description of the intervention(s) and comparator(s), including intervention name, treatment dose, duration, components and how it was delivered.
▸ Intervention type (eg, pharmaceutical intervention, traditional Chinese medicine, non-pharmacological treatment and multicomponent interventions).

Outcomes
▸ Outcome name including the name of each outcome of interest and how it was measured (instruments used).
▸ Outcome type and reported format. The components of reporting effect measures are: (1) the effect measure itself (eg, change from baseline), (2) a measure of its variance (eg, the SD or the 95% CI), (3) the number of participants in the study arm (N).
▸ Scale and direction of effect.

Results
▸ Results of outcomes reported in the original study at each time point.

Risk of bias (RoB) information
▸ Judgements based on the criteria of the RoB 2.

---

EUROHIS-QOL 8-item index and the Zarit Burden Interview, respectively.

If the outcome of interest or its measure is not reported in included studies, we will extract the outcome results that are reported as the primary outcome in the original included study. Where feasible, we will also be open to examining other outcomes evaluated in the included studies.

We will extract results of outcomes of interest measured at each time point reported in included studies. Nevertheless, we will afford preference to the endpoint of the study in the main data synthesis. Results at multiple time points will be used for subgroup analysis and meta-regression to explore the short-term and long-term effects of outcomes.

### RoB in individual studies

We will use the Cochrane Collaboration's recently updated RoB tool[31] to assess the quality of included studies in Covidence. Two reviewers will make independent judgements based on the criteria for judging the RoB. Disagreement will be resolved by discussion and arbitrated by a third reviewer if consensus is unreachable. For studies that do not provide sufficient information in full texts for RoB assessment, we will search for the study's protocol, trial registry information or other relevant materials to facilitate the judgement. The absence of a prespecified analysis plan may raise some concerns in the domain for bias in selection of the reported result.

### Data synthesis

Evidence on dementia-related interventions in PLwDs and carers will be analysed separately. Studies of family-based or dyadic interventions involving both PLwDs and carers will be categorised according to the subject of each outcome.

### Narrative synthesis

To map dementia-related interventions conducted in Chinese communities, we will undertake a narrative synthesis to fully interpret the extracted evidence from all included studies. We will first describe and summarise disease characteristics, features of the intervention, number of participants, participant characteristics, outcomes, outcome measures and indication of RoB assessment in a tabular form. In line with the Guidance on the Conduct of Narrative Synthesis in Systematic Reviews,[40] we will then explore the relationship among types of interventions (or details of pharmacological, non-pharmacological and multicomponent interventions), outcomes and outcome measures conducted in Chinese populations. Idea webbing will be used to visually describe conceptual linkages through examination of extracted data if feasible. The key questions here are what (types of) interventions have been conducted in Chinese communities, what specific outcomes those interventions target and what measures are used for those outcomes. We expect to identify research gaps in this field for future studies and practices.

### Meta-analyses

To compare the effectiveness of interventions for outcomes of interest (described in the Outcomes and prioritisation section), we will conduct quantitative synthesis of treatment effects through meta-analyses where sufficient information is available. For a specific outcome, we will perform a series of pairwise meta-analyses for all direct comparisons (eg, one comparison between an intervention group and a control or another intervention group).[41] Due to the underlying difference between studies in terms of participants, intervention details and care settings, a random-effects pooling model will be conducted by default for an overall summary estimate by weighting studies using a combination of within-study and between-study variance. When the included studies use different instruments to evaluate the same outcome (eg, MMSE, MoCA and ADAS-Cog for measuring cognition), we will use standardised mean difference (the absolute mean difference between the intervention group and control group divided by the SD in the control group) for continuous outcomes and relative risks for dichotomous outcomes to compute the effect size for each study.

To compare the effectiveness for multiple interventions, we will use network meta-analysis to combine direct and indirect evidence for relevant treatment effects.[42] In network meta-analyses, different comparisons among two or more of the treatments can be included in one analysis. We will generate network geometry to visualise and assess the treatment networks and estimate and combine comparative effects from direct and indirect evidence. In examining the transitivity hypothesis of network meta-analysis, we will use 'loop-specific approaches' to detect the inconsistency of a network of interventions, including local inconsistency test to evaluate the loop inconsistency in regions of network separately[43] and global inconsistency test to evaluate the incoherence in the overall network.[44]

Sensitivity analyses will be conducted to explore the robustness of the meta-analysis results by varying the analytic data or methods, including analysing studies only with a low RoB and trials using a placebo as a comparator.[45]

### Dealing with missing data

When there are missing data, we will attempt to obtain these by contacting the study author(s). If unsuccessful, we will consider using imputation methods to impute the missing value[46] or exclude studies with missing data from the quantitative analysis. We will use sensitivity analysis to evaluate the potential influence on the overall treatment effects of included studies that use per-protocol analysis or suggest that the result was biased by missing outcome data (ie, high RoB) based on the RoB 2 assessment tool.[31]

### Subgroup analysis and meta-regression

We will calculate Cochrane's Q statistic and the $I^2$ statistic to estimate the heterogeneity of the included studies.[47] If statistical heterogeneity is observed, we will conduct subgroup analysis and meta-regression to explore the

potential reasons for the differences. Potential candidate covariates for subgroup analysis and meta-regression include intervention characteristics (eg, types of intervention, intervention dosage and duration), participant characteristics (eg, age, gender, education, severity of dementia and type of dementia), care settings, follow-up period (eg, at 3, 6 and 12 months) and locations (eg, mainland China, Hong Kong, Taiwan, Macau and other Chinese communities worldwide).

## Meta-bias(es)

For each meta-analysis, we will use a funnel plot asymmetry assessment to detect meta-biases. Statistical tests for funnel plot asymmetry will be performed when at least 10 studies included in the meta-analysis.[48] Contour lines indicating various statistical significance will be used to aid visual interpretation of funnel plots. If funnel plot asymmetry is observed, we will also consider other possible reasons apart from non-reporting bias such as poor methodological quality and true heterogeneity of the included studies.[49]

## Confidence in cumulative estimate

We will use the Grading of Recommendations Assessment, Development and Evaluation approach[50] to assess the quality of evidence. The domains of the assessment include RoB, inconsistency, indirectness of evidence, imprecision and publication bias.

## Patient and public involvement

Neither patients nor the public will be involved in the design or development of this review protocol. However, stakeholders, including PLwD, family members, care staff, healthcare professionals and policymakers, will be engaged in the dissemination plan as described below.

## ETHICS AND DISSEMINATION

This protocol for a systematic review and meta-analysis describes the methods to identify and synthesise published evidence on the effectiveness of interventions for PLwD and their carers in Chinese communities. No formal ethics approval is required for this protocol. The findings from this study will facilitate the development of studies on interventions for dementia and provide timely information for dementia policymaking and practice. We will target both professionals and non-specialist audiences in disseminating the outcomes of the review through prints and events, including peer-reviewed publications, conference presentations, public events, and publicly accessible websites.

### Author affiliations
[1]Department of Social Work and Social Administration, The University of Hong Kong, Hong Kong SAR, China
[2]Center for Social Welfare Studies, Beijing Normal University, Beijing, China
[3]Care Policy and Evaluation Centre (CPEC), London School of Economics and Political Science, London, UK
[4]Faculty of Social Sciences, The University of Hong Kong, Hong Kong SAR, China

**Contributors** GW, CS, SC, JCPC, AC-H and MK defined the scope of the review and review question. GW, CS, SC and MS-K developed inclusion/exclusion criteria. HL designed the method for the Chinese bibliographical database search. GW, CS, SC, MS-K and DMD consulted on the bibliographical databases to be searched and search terms used. GW, CS, MS-K, SC, JCPC and DKYL provided information on methods of data synthesis. CS drafted the protocol. CS, SC, MS-K, JCPC, HL, DKYL, XC, YZ, RD, AC-H, DMD, MK and GW contributed to the study design and reviewed the draft protocol. GW is the guarantor of the manuscript.

**Funding** This work is conducted as part of the 'Tools to Inform Policy: Chinese Communities Actions in Response to Dementia' (TIP-CARD) project, supported by the Hong Kong Research Impact Fund of the Research Grants Council (Project Reference Number: R7017-18). MS-K, DMD, MK and AC-H's contributions were supported by the UK Research and Innovation's Global Challenges Research Fund (ES/P010938/1) as part of the 'Strengthening Responses to Dementia in Developing Countries' (STRiDE) project.

**Disclaimer** The funders were not involved in the development of this protocol.

**Competing interests** None declared.

**Patient and public involvement** Patients and/or the public were not involved in the design, or conduct, or reporting, or dissemination plans of this research. Refer to the Methods section for further details.

**Patient consent for publication** Not required.

**Provenance and peer review** Not commissioned; externally peer reviewed.

**ORCID iDs**
Cheng Shi http://orcid.org/0000-0002-7341-3135
Maximilian Salcher-Konrad http://orcid.org/0000-0002-5628-5266
Hao Luo http://orcid.org/0000-0003-4261-3414
Dara Kiu Yi Leung http://orcid.org/0000-0002-7255-2790
Gloria Wong http://orcid.org/0000-0002-1331-942X

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
