## [Reviewer comments · BMJ Open]

ARTICLE DETAILS

TITLE (PROVISIONAL)	Effectiveness of interventions for people living with dementia and their carers in Chinese communities: protocol for a systematic review and meta-analysis of randomised controlled trials
AUTHORS	Shi, Cheng; Chen, Shuangzhou; Salcher-Konrad, Maximilian; Choy, Jacky CP; Luo, Hao; Leung, Dara Kiu Yi; Cai, Xinxin; Zeng, Yue; Dai, Ruizhi; Comas-Herrera, Adelina; McDaid, David; Knapp, Martin; Wong, Gloria

VERSION 1 – REVIEW

REVIEWER	Trieu, Calvin VU University Medical Centre Amsterdam, Alzheimer Center
REVIEW RETURNED	08-Feb-2021

GENERAL COMMENTS	Page 5 line 19: "A wide range of pharmacological and non-pharmacological interventions for dementia have been trialled in Chinese populations, although the evidence has not been systemically reviewed." Incorrect sentence structure. Page 8 line 4: "Do you mean scarce instead of sparse" Page 8 line 26-27: Evidence for what? Page 10 line 14-18: " Dementia or major...neurocognitive disorder." Unclear what's meant by this sentence. Page 11 line 21: What outcomes will be used concretely for each type of dementia. Will inventories or questionnaires (e.g. NPI, FBI, SRI) be used as measuring outcome for cognition. And how will you compare these between studies? I advise the authors to find a native English speaker to proofread the manuscript.
---

REVIEWER	Ma, Kris Pui Kwan University of Washington
REVIEW RETURNED	05-Mar-2021

GENERAL COMMENTS	BMJ Open – Peer Review Manuscript ID: bmjopen-2020-047560 This submission described a protocol of a systematic review and meta-analysis that evaluates the evidence of pharmacological and non-pharmacological interventions for persons living with dementia (PLWD) and carers in Chinese populations worldwide. The strengths of this study include addressing a major research gap in the literature of dementia care where Chinese populations are underrepresented, planning for a comprehensive search of Chinese and English publications, and an inclusive review of both pharmacological and non-pharmacological interventions. The manuscript is well written and organized. However, several
--

	concerns and weaknesses keep the study from realizing its potential; most significantly, this protocol did not include a well-defined scientific question that informs the population and outcomes on which the review will focus. My biggest concern of this protocol is the lack of a targeted research question that this review intends to address. Although the overall project aims to synthesize and evaluate the evidence of interventions for PLWD and carers in Chinese populations, there are many directions that a single review like this can take. As the authors alluded in the introduction, this review can compare evidence of pharmacological and non-pharmacological interventions in certain outcomes among Chinese populations, or compare evidence of interventions between Chinese PLWD and carers living in Chinese societies versus western societies due to cultural differences, or evaluate the evidence of interventions by types of dementia in Chinese populations. It is unclear from the protocol what direction this review is taking, thus leading to poorly defined PICO (Population-Intervention-Comparison-Outcomes) for this review: (1) Population Be more specific in the inclusion/exclusion criteria for dementia carers. When the authors say “include everyone providing care”, it is unclear whether this also includes indirect carers – those who have no direct regular contact with PLWD but still contribute to the care experience of PLWD, such as nursing managers or administrative higher-ups who run long-term care facilities? Or family members who do not provide daily assistance but help make healthcare decisions for PLWD? Provide more information for inclusion/exclusion criteria for PLWD, such as gender, age of diagnosis, and care settings (home, long-term care facilities, hospitals) (2) Outcomes Clearly define and specify the outcomes of interest can help determine the appropriateness of the interventions to be included in this review and potentially inform how this review can contribute to the existing literature. When the review has a focused research question, the authors can define a series of core outcome sets (e.g. clinical outcomes, health service utilization outcomes, caregiver-related outcomes, caregiver-PLWD dyadic outcomes, society/system change outcomes) to be included in the review and still be comprehensive and inclusive of important parts of the evidence base for dementia interventions related to the research question. Outcomes and prioritization – Although the authors did not specify the outcomes of interest, the authors later noted that they will prioritize evaluating PLWD’s cognition, functioning and caregivers’ quality of life. Provide justification of why cognition, functioning and quality of life are the primary outcomes of interest. Does the review include evaluating short-term and long-term outcomes? My second concern is the insufficient details in the protocol that allow others to replicate the review:
--	--

	(1) Population – In some studies conducted in the U.S., U.K., Canada, Australia or other western countries, they included Asians in their sample but did not specify the Asian subgroups or ethnicities; Chinese might be included in these studies. Provide a plan to address this issue. Also, some studies did not specifically report the types of dementia for their sample, please provide a plan on how this issue may be addressed if it is relevant to the review. (2) Study selection – Are the researchers who screen and review the studies bilingual and can read English and Chinese, given the inclusion of Chinese and English publications? (3) Data extraction – Provide a list of data that will be extracted. (4) Data extraction – Provide a plan on how the data will be extracted from Chinese and English publications. Will the data be extracted in Chinese or English? Who and how to translate Chinese extracted data into English (or vice versa)? Provide a plan to ensure the translation is accurate. (5) Risk of bias – Provide a plan on what will be done when there is insufficient information from an article to assess for bias. (6) Data synthesis – Provide more information on what study data will be qualitatively synthesized and what study data will be quantitatively synthesized and how? (7) Qualitative analysis – Provide a justification of why the method of narrative synthesis is chosen. (8) Quantitative synthesis – Provide a plan on how missing data will be dealt with. (9) Dissemination – Provide more details on how stakeholders such as PLWD, family members, care staff, healthcare professionals and policymakers will engage in the planned dissemination channels (peer-reviewed publication, conference presentations, and publicly accessible websites). Other suggested revisions include: (1) Study design – “To minimize small study-effects, we will exclude the studies with a sample size of less than 50 in either the intervention group or comparison group(s)” – It is unclear of whether the sample size is referring to the total sample size of the group (whether it is Chinese and other racial/ethnic groups) or just Chinese participants in the group? (2) Qualitative analysis – The heading “Qualitative analysis” is confusing. I recommend using “Narrative synthesis” as the heading, because what the authors propose to do is to generate a summary of study, intervention and participant characteristics in a narrative and tabular format.
--	---

VERSION 1 – AUTHOR RESPONSE

Reviewer: 1

Dr. Calvin Trieu, VU University Medical Centre Amsterdam

Comments to the Author:	Author responses
1. Page 5 line 19: "A wide range of pharmacological and non-pharmacological interventions for dementia have been trialled in Chinese populations, although the evidence has not been systemically reviewed." Incorrect sentence structure.	Thank you for this comment. To reduce the confusion, we have changed the conjunction. In our revised manuscript, the revised sentence is "Although a wide range of pharmacological and non-pharmacological interventions for dementia have been trialled in Chinese populations, the evidence has not been systemically synthesised" (p3, para1).
2. Page 8 line 4: "Do you mean scarce instead of sparse"	Thanks for your suggestion. The word 'scarce' should be more precise in this sentence. We have made this change (p6, para 1).
3. Page 8 line 26-27: Evidence for what?	Thanks for your comments. We have expanded the sentence as "...evidence on the effectiveness of dementia-related interventions relevant to local population" (p6, para 1).
4. Page 10 line 14-18: " Dementia or major...neurocognitive disorder." Unclear what's meant by this sentence.	Thank you for this query. To address this, we have made a substantial revision on the paragraph for explaining the eligible population in this systematic review. Clarifications on types of dementia and mild cognitive impairment have been added in eligible criteria (Methods and analysis – Eligibility criteria- Population; p 9, para 4): "Dementia, as a major neurocognitive disorder, describes a group of symptoms of cognitive decline, including, but not limited to, Alzheimer's disease, vascular dementia, frontotemporal dementia, Lewy body disease, and mixed dementia. Studies conducted among people living with MCI, mild neurocognitive disorder, vascular cognitive impairment, and no dementia (VCIND) will be eligible for inclusion due to the higher risk of developing dementia in later years.[1]"
5. Page 11 line 21: What outcomes will be used concretely for each type of dementia. Will inventories or questionnaires (e.g. NPI, FBI, SRI) be used as measuring outcome for cognition. And how will you compare these between studies?	Thanks for this query. As one of our research aims is to identify and map out the dementia-related intervention trialled in Chinese communities, which is an exploratory research question in nature, we accept any type of outcome from the perspective of effectiveness for study selection. To specify this, we have added potential outcomes measured in existing evidence on effectiveness of dementia-related intervention according to a dementia evidence toolkit (https://www.modem-

	dementia.org.uk/) and mentioned we accept all types of outcome measurements(p11, para3): “According to the MODEM dementia evidence toolkit (https://www.modem-dementia.org.uk/), outcomes measured in existing studies may include 1) cognition, behavioural and psychological symptoms, functional status, physical health and quality of life of PLwD; 2) carer burden[2-4] , the carer’s mental health, quality of life and other carer outcomes (e.g., financial burdens); 3) service use, cost reduction (including hospital use reduction and care home admission delay) and service satisfaction; 4) risk reduction (of dementia and co-morbidities) and prevention or management of comorbidities. To capture the diversity of interventions trialled in Chinese communities, we will accept all outcome measures that reflect intervention effectiveness.” Regarding the instruments for measuring the outcome of our interests in data extraction and analysis, we have added some scales or assessments which are commonly used for our primary outcomes and secondary outcomes for PLwD and their carers separately (p18 para1-2): “As dementia is a condition affecting cognition by definition, our primary outcome of interest in studies conducted among PLwD will be changes in cognition. Common assessments for measuring cognitive impairment level or performance include the Mini–Mental State Examination (MMSE), the Montreal Cognitive Assessment (MoCA), and the Alzheimer's Disease Assessment Scale-Cognitive Subscale (ADAS-Cog). Given the effects of dementia on the ability to organise activities,[5] we will also focus on changes in functional ability following treatment. For example, the Disability Assessment for Dementia (DAD) is designed for evaluating functional ability to complete activities of daily living (ADL) and instrumental activities of daily living (IADLs) among people living with dementia. Caring for a person living with dementia can be very stressful, which may lead to a higher level of depression or health issues.[6] For studies conducted among carers of PLwD, we will focus on changes in quality of life and carer burden[2,3], as measured by tools such as the EUROHIS-QOL 8-item index and the Zarit Burden Interview (ZBI), respectively.”
--	--

	To compare the effectiveness of interventions for outcomes of interest (described in the ‘Outcome and prioritisation’ section), we will conduct quantitative synthesis of treatment effects through meta-analyses where sufficient information is available. For a specific outcome, we will perform a series of pairwise meta-analyses for all direct comparisons (e.g., one comparison between an intervention group and a control or another intervention group).[7] When the included studies use different instruments to evaluate the same outcome (e.g., MMSE, MoCA, and ADAS-Cog for measuring cognition), we will use standardised mean difference (the absolute mean difference between the intervention group and control group divided by the standard deviation in the control group) for continuous outcomes and relative risks for dichotomous outcomes to compute the effect size for each study. We presented those in our meta-analysis plan (p20, para 2; p21 para 2).
6. I advise the authors to find a native English speaker to proofread the manuscript.	Thank you for this suggestion. The revised manuscript has been proofread by a professional editor.

Reviewer: 2

Dr. Kris Pui Kwan Ma, University of Washington

Comments to the Author:	Author responses
1. This submission described a protocol of a systematic review and meta-analysis that evaluates the evidence of pharmacological and non-pharmacological interventions for persons living with dementia (PLWD) and carers in Chinese populations worldwide. The strengths of this study include addressing a major research gap in the literature of dementia care where Chinese populations are underrepresented, planning for a comprehensive search of Chinese and English publications, and an inclusive review of both pharmacological and non-pharmacological interventions. The manuscript is well written and organized.	Thank you for recognising the strengths of the study.
2. However, several concerns and weaknesses keep the study from	Thank you very much for these informative and constructive comments. We set out our point-by-

realizing its potential; most significantly, this protocol did not include a well defined scientific question that informs the population and outcomes on which the review will focus.	point response below. The changes we have made to the manuscript are highlighted in track changes mode.
3. My biggest concern of this protocol is the lack of a targeted research question that this review intends to address. Although the overall project aims to synthesize and evaluate the evidence of interventions for PLWD and carers in Chinese populations, there are many directions that a single review like this can take. As the authors alluded in the introduction, this review can compare evidence of pharmacological and non-pharmacological interventions in certain outcomes among Chinese populations, or compare evidence of interventions between Chinese PLWD and carers living in Chinese societies versus western societies due to cultural differences, or evaluate the evidence of interventions by types of dementia in Chinese populations.	Thank you for raising this important point. To address this, we have further elaborated our two research aims in the introduction (p8, para2), which are: 1) map out interventions for dementia studied in Chinese communities, and 2) compare the effectiveness of those interventions for achieving desired outcomes. To capture the diversity of interventions, we attempt to include related studies regardless of types of outcomes they used. The perspective of effectiveness[8,9] is used to identify any dementia-related interventions trialled in Chinese populations. We also agree that there may be many directions in potential eligible studies, and it is impossible to compare all intervention outcomes. Therefore, we prioritised outcomes for data extraction and analysis when comparing the effectiveness of interventions. Details are provided in response to the comments on outcomes below (4. (2)).
4. It is unclear from the protocol what direction this review is taking, thus leading to poorly defined PICO (Population-Intervention-Comparison-Outcomes) for this review:	Thanks for your suggestion. We have modified each part of the PICO according to your comments. Point-by-point responses to the PICO are provided below.
(1) Population a. Be more specific in the inclusion/exclusion criteria for dementia carers. When the authors say “include everyone providing care”, it is unclear whether this also includes indirect carers – those who have no direct regular contact with PLWD but still contribute to the care experience of PLWD, such as nursing managers or administrative higher-ups who run long-term care facilities? Or family members who do not provide daily assistance but help make healthcare decisions for PLWD?	Thank you for this comment. Yes. Any type of carers as long as they have been involved in dementia-related care provision and management, will be acceptable. Accordingly, we have elaborated the eligibility for population in p10, para2: “Our definition of dementia carer refers to persons involved in care provision and management and will not depend on whether or not the carer is paid, lives with the person they care for, or provide direct or indirect care. Therefore, dementia carers can refer to health and social care professionals, care managers, care workers, administrative staff of care facilitators, family carers, other unpaid carers, and family members assisting with care decisions.”

b. Provide more information for inclusion/exclusion criteria for PLWD, such as gender, age of diagnosis, and care settings (home, long-term care facilities, hospitals)	Thank you for this comment. We have further clarified eligible criteria for population. As we aim to identify dementia-related interventions in Chinese communities, gender, age of diagnosis and care setting will not be used for excluding studies. While in data collection, we will extract data on gender and care settings in original studies for analysis. We have added two sentences on defining the eligible criteria on population (p 9, para 3): “We will include relevant studies conducted in any type of care settings, such as home, community, residential homes, clinics, hospitals, and other care settings. Participants characteristics such as gender, education, and age at diagnosis will not be used for excluding studies.”
(2) Outcomes a. Clearly define and specify the outcomes of interest can help determine the appropriateness of the interventions to be included in this review and potentially inform how this review can contribute to the existing literature. When the review has a focused research question, the authors can define a series of core outcome sets (e.g. clinical outcomes, health service utilization outcomes, caregiver-related outcomes, caregiver-PLWD dyadic outcomes, society/system change outcomes) to be included in the review and still be comprehensive and inclusive of important parts of the evidence base for dementia interventions related to the research question.	Thanks for this suggestion. Indeed, the protocol does not specify in details on outcome of interest in study selection. The rationale behind is that we aim to identify and map out the dementia-related intervention trialled in Chinese communities, which is an exploratory research question. We have highlighted the research aim in revised introduction (p8, para2). Given this, we accept any types of outcome from the perspective of effectiveness when including studies. To specify this, we have added potential outcomes measured in exiting evidence on effectiveness of dementia-related intervention according to a dementia evidence toolkit (https://www.modem-dementia.org.uk/; p11, para3): “Any type of outcomes of dementia-related intervention will be eligible for inclusion from the perspective of effectiveness, which may affect individuals, families, the dementia care workforce, wider society, and social or health care systems. Dementia often triggers complex problems in many domains.[5] According to the MODEM dementia evidence toolkit (https://www.modem-dementia.org.uk/), outcomes measured in existing studies may include 1) cognition, behavioural and psychological symptoms, functional status, physical health and quality of life of PLwD; 2) carer burden, the carer’s mental health, quality of life and other carer outcomes (e.g., financial burdens); 3) service use, cost reduction (including hospital use reduction and care home admission delay) and service satisfaction; 4) risk reduction (of demetia and co-morbidities) and prevention or management of comorbidities. To capture the diversity of interventions trialled in Chinese communities, we will accept all outcome measures that reflect intervention effectiveness. ”

	We agree with the reviewer on the importance of defining clear outcomes when evaluating and comparing the effectiveness of interventions. We have now made substantial revisions in the 'Outcomes and prioritisation' section (p17, para 3; p18 para1-4) to prioritise outcomes of intervention for our comparative analysis. Further clarifications are provided in the next response.
b. Outcomes and prioritization – Although the authors did not specify the outcomes of interest, the authors later noted that they will prioritize evaluating PLWD's cognition, functioning and caregivers' quality of life. Provide justification of why cognition, functioning and quality of life are the primary outcomes of interest.	Thank you for this comment. Following the last response, we will accept any outcomes in study selection aiming to map out the interventions studied in Chinese communities. For comparing the effectiveness of interventions, we prioritise outcomes of interests to make it more feasible. As acknowledged by the reviewer, there can be many directions in terms of outcomes. Given the broad range of potentially relevant outcomes, it can be very challenging to compare all the outcomes. To address this confusion, we have added a paragraph to specify this point (p17, para3). The reasons of why we prioritise cognition and functional status for studies on PLwD and quality of life and carer burden for studies on dementia carer have been laid out in p16, para 4: “As dementia is a condition affecting cognition by definition, our primary outcome of interest in studies conducted among PLwD will be changes in cognition. Common assessments for measuring cognitive impairment level or performance include the Mini–Mental State Examination (MMSE), the Montreal Cognitive Assessment (MoCA), and the Alzheimer's Disease Assessment Scale–Cognitive Subscale (ADAS-Cog). Given the effects of dementia on the ability to organise activities,[5] we will also focus on changes in functional ability following treatment. For example, the Disability Assessment for Dementia (DAD) is designed for evaluating functional ability to complete activities of daily living (ADL) and instrumental activities of daily living (IADLs) among people living with dementia. Caring for a person living with dementia can be very stressful, which may lead to a higher level of depression or health issues.[6] For studies conducted among carers of PLwD, we will focus on changes in quality of life and carer burden[2,3], as measured by tools such as the EUROHIS-QOL 8-item index and the Zarit Burden Interview (ZBI), respectively.”

	In addition, we also considered the situation that no outcome of interest or its measure is reported in included studies. If so, we will extract the outcome results that is reported as the primary outcome of the original study (p18, pare 3).
c. Does the review include evaluating short-term and long-term outcomes?	Thank you for this comment. Yes. We expect to evaluate the short-term and long-term outcomes if sufficient information is provided. Therefore, the revised manuscript laid out that we will extract results of primary and secondary outcomes measured at each time point reported in included studies, although we will give preference for the endpoint of the study in main data synthesis (p18, para4). The short-term and long-term effect of outcomes will be evaluated in subgroup analysis and meta-regression if feasible. We have added the explanation in the revised manuscript (p18, para 4; p22, para 3).
5. My second concern is the insufficient details in the protocol that allow others to replicate the review:	Thanks for your comments. Point-by-point responses to the issues raised by the reviewer are provided below.
(1) Population – In some studies conducted in the U.S., U.K., Canada, Australia or other western countries, they included Asians in their sample but did not specify the Asian subgroups or ethnicities; Chinese might be included in these studies. Provide a plan to address this issue.	Thanks for this suggestion. To address this issue, we have elaborated that Studies without explicating the proportion of Chinese participants over 50% or studies without a specific subgroup analysis for Chinese participants will be excluded. (p10, para3).
Also, some studies did not specifically report the types of dementia for their sample, please provide a plan on how this issue may be addressed if it is relevant to the review.	Thanks for this comment. For this point, the revised manuscript has highlighted studies not specifying the type of dementia are acceptable for inclusion (p 10, para 1). While in subgroup analysis, we may exclude those studies without reporting the type of dementia depending on the characteristics of studies included eventually.
(2) Study selection – Are the researchers who screen and review the studies bilingual and can read English and Chinese, given the inclusion of Chinese and English publications?	Thanks for this query. Yes. Our reviewers for title and abstract screening and full-text review can read both English and Chinese well. We have clarified this point in study selection(p16, para3): “Reviewers for title and abstract screening and full-text review will be able to read and understand inclusion/exclusion criteria for publications in both English and simplified Chinese. ”
(3) Data extraction – Provide a list of data that will be extracted.	Thanks for this comment. We have substantially fleshed out the list of data to be extracted from included studies in Box 1 (p34):

	General information  • Reviewers' name • Date of data extraction • Publication details and identification • Sponsorship source • Research site: places (city-level) where the trial was conducted • Setting (e.g., hospital, care home, community) • Study aim(s) • Publication language: Chinese or English Methods  • Study design • List of all outcomes with instruments reported in study Population  • Inclusion criteria • Exclusion criteria • Group differences • Clinical features (e.g., types of dementia, severity and duration of dementia) • Baseline characteristics of participants in each study arm or overall participants: demographics (e.g., age, gender), socio-economic status (e.g., education), clinical outcomes if any, number of participants Intervention  • Description of the intervention(s) and comparator(s) (e.g., name, treatment dose, duration, components and how it was delivered). • Intervention type (e.g., pharmaceutical intervention, traditional Chinese medicine, non-pharmacological treatment and multicomponent interventions). Outcomes  • Outcome name including name of each outcome of interest and how it was measured (instruments used) • Outcome type and reported format. The components of reporting effect measures are: (1) the effect measure itself (e.g., change from baseline), (2) a measure of its variance (e.g., the standard deviation, or the 95% CI), (3) the number of participants in the study arm (N). • Scale and direction of effect Results  • Results of outcomes reported in the original study at each time point. Risk of bias information  • Judgements based on the criteria of the RoB 2.
--	--

(4) Data extraction – Provide a plan on how the data will be extracted from Chinese and English publications. Will the data be extracted in Chinese or English? Who and how to translate Chinese extracted data into English (or vice versa)? Provide a plan to ensure the translation is accurate.	Thanks for pointing this out. Our data extraction form is in English. When extracting data from publications in Chinese, reviewers can complete the extraction form by using the original expression in Chinese, except for outcome name and brief introduction of intervention which need to be recorded in English. We will use the English abstract if available or translate that information by reviewers. The final results from both English and Chinese studies will be verified by one bilingual researcher to ensure a consistent translation. Those points have been updated in “Data collection process” section (p16, para 5)
(5) Risk of bias – Provide a plan on what will be done when there is insufficient information from an article to assess for bias.	Thanks for your suggestion. For studies that do not provide sufficient information in full-texts for RoB assessment, we will search for the study’s protocol, trial registry information, or other relevant materials to facilitate the judgement. The absence of a pre-specified analysis plan may raise some concerns in the domain for bias in selection of the reported result. Those points have been added in p 19, para 1.
(6) Data synthesis – Provide more information on what study data will be qualitatively synthesized and what study data will be quantitatively synthesized and how?	Thanks for this comment. In the revised manuscript, we have expanded data synthesis plan and specified how extracted data will be qualitatively and quantitatively synthesised based on our research questions. The modified plan specified that all extracted data will be narratively (qualitatively) synthesised to map out interventions conducted in Chinese communities. The points have been added in p 19, para 3: “To map dementia-related interventions conducted in Chinese communities, we will undertake a narrative synthesis to fully interpret the extracted evidence from all included studies. We will first describe and summarise disease characteristics, features of the intervention, number of participants, participant characteristics, outcomes, outcome measures, and indication of risk of bias assessment in a tabular form. In line with the Guidance on the Conduct of Narrative Synthesis in Systematic Review,[10] we will then explore the relationship among types of interventions (or details of pharmacological, non-pharmacological and multicomponent interventions), outcomes, and outcome measures conducted in Chinese populations. Idea webbing will be used to visually describe conceptual linkages through examination of extracted data if feasible. The key questions here are what (types of) interventions have been conducted in Chinese communities, what specific outcomes those interventions target and

	what measures are used for those outcomes. We expect to identify research gaps in this field for future studies and practice.” In the meta-analysis (quantitatively synthesis), we will synthesise evidence on the effectiveness of interventions for outcomes of interests presented in “Outcome and prioritisation” section for pairwise and network meta-analysis. The aim of quantitative analysis is to evaluate and compare the effectiveness of interventions. However, due to anticipated diversity of outcomes, the comparison of effectiveness will focus on the outcomes of interests. Those points are reflected in p 20 para 2 and p 21 para 1: “To compare the effectiveness of interventions for outcomes of interest (described in the ‘Outcome and prioritisation’ section), we will conduct quantitative synthesis of treatment effects through meta-analyses where sufficient information is available. For a specific outcome, we will perform a series of pairwise meta-analyses for all direct comparisons (e.g., one comparison between an intervention group and a control or another intervention group).[7] Due to the underlying difference between studies in terms of participants, intervention details and care settings, a random-effects pooling model will be conducted by default for an overall summary estimate by weighting studies using a combination of within- and between-study variance. When the included studies use different instruments to evaluate the same outcome (e.g., MMSE, MoCA, and ADAS-Cog for measuring cognition), we will use standardised mean difference (the absolute mean difference between the intervention group and control group divided by the standard deviation in the control group) for continuous outcomes and relative risks for dichotomous outcomes to compute the effect size for each study. To compare the effectiveness for multiple interventions, we will use network meta-analysis to combine direct and indirect evidence for relevant treatment effects.[11] In network meta-analyses, different comparisons among two or more of the treatments can be included in one analysis. We will generate network geometry to visualise and assess the treatment networks, and estimate and combine comparative effects from direct and indirect evidence. In examining the transitivity hypothesis of network meta-analysis, we will use ‘loop-specific approaches’ to detect the inconsistency of a network of interventions, including local inconsistency test to
--	---

	evaluate the loop inconsistency in regions of network separately[12] and global inconsistency test to evaluate the incoherence in the overall network.[13] ”
(7) Qualitative analysis – Provide a justification of why the method of narrative synthesis is chosen.	Thank you for this suggestion. Following the last response, we have now expanded the paragraph on Narrative synthesis to elaborate the purpose of using this method and how it will be conducted. Based on a previous project(“MODEM”) and an ongoing project(“STRiDE”) which explored the dementia-related interventions in high-income settings and low-and-middle-income settings using English bibliographical databases, we anticipate there will be an increasing number of evidence with diverse interventions and outcomes. Therefore, all our extracted data may not be suitable for combining quantitatively. To map dementia-related interventions conducted in Chinese communities, we plan to use narrative synthesise to fully interpret all extracted data to answer what interventions have been conducted, whether those interventions targeted specific outcomes and what measures used for those outcomes in Chinese communities. We will follow the Guidance on the Conduct of Narrative Synthesis in systematic review[14] to explore the relationship among interventions, outcomes and measures conducted in Chinese communities. To reflecting those point, we substantially revised the paragraph for narrative synthesis (p19, para 3): “To map out dementia-related interventions conducted in Chinese communities, we will undertake a narrative synthesis to fully interpret the extracted evidence from all included studies. We will first describe and summarise disease characteristics, features of the intervention, number of participants, participant characteristics, outcomes, outcome measures, and indication of risk of bias assessment in a tabular form. In line with the Guidance on the Conduct of Narrative Synthesis in Systematic Review[14], we will explore the relationship among types of interventions, outcomes and outcome measures conducted in Chinese populations. The key question here is what (types of) interventions have been conducted in Chinese communities, whether those interventions targeted specific outcomes, and what measures were used for those outcomes. We expect to identify the research progress and research gaps in this field to inform future direction of study and practice.”

(8) Quantitative synthesis – Provide a plan on how missing data will be dealt with.	Thanks for this suggestion. We have added a paragraph on dealing with missing data in our revised manuscript under Data synthesis (p 22, para 2) : “When there are missing data, we will attempt to obtain these by contacting the study author(s). If unsuccessful, we will consider using imputation methods to impute the missing value[15] or exclude studies with missing data from the quantitative analysis. We will use sensitivity analysis to evaluate the potential influence on the overall treatment effects of included studies that use per-protocol analysis or suggest the result was biased by missing outcome data (i.e., high risk of bias) based on the RoB 2 assessment tool.[16]”
(9) Dissemination – Provide more details on how stakeholders such as PLWD, family members, care staff, healthcare professionals and policymakers will engage in the planned dissemination channels (peer reviewed publication, conference presentations, and publicly accessible websites).	Thanks for this comment. To address this, we have highlighted we will target both professionals and non-specialist audiences in disseminating the outcomes of the review through print and events, including peer-reviewed publications, conference presentations, public events, and publicly accessible websites. The relevant points are reflected “Ethics and dissemination” section (p24, para 1) and Abstract (p4, para2).
6. Other suggested revisions include: (1) Study design – “To minimize small study-effects, we will exclude the studies with a sample size of less than 50 in either the intervention group or comparison group(s)” – It is unclear of whether the sample size is referring to the total sample size of the group (whether it is Chinese and other racial/ethnic groups) or just Chinese participants in the group?	Thank you for this comment. To clarify this point, we have specified that for studies conducted with a population of mixed ethnicity, the sample size of each study arm for Chinese sub-group analysis should be greater than 50 participants; for studies in which more than 50% of participants are Chinese and all participants are randomly grouped, the sample size of each study arm is expected to be greater than 50 participants regardless of ethnicity (p 12, para 2).
(2) Qualitative analysis – The heading “Qualitative analysis” is confusing. I recommend using “Narrative synthesis” as the heading, because what the authors propose to do is to generate a summary of study, intervention and participant characteristics in a narrative and tabular format.	Thank you for this suggestion. We agree on this point. To reduce the confusion, we have changed the subheading “Qualitative analysis” to “Narrative synthesis” and also changed “Quantitative analysis” to “Meta-analysis” to make the content more straightforward (p19; p20).

References

1. Mitchell AJ, Shiri-Feshki M. Rate of progression of mild cognitive impairment to dementia—meta-analysis of 41 robust inception cohort studies. *Acta Psychiatrica Scandinavica* 2009;119(4):252-65.
2. Chien WT, Lee IYM. Randomized controlled trial of a dementia care programme for families of home-resided older people with dementia. *J Adv Nurs* 2011;67(4):774-87. doi: 10.1111/j.1365-2648.2010.05537.x
3. Lam LCW, Lee JSW, Chung JCC, et al. A randomized controlled trial to examine the effectiveness of case management model for community dwelling older persons with mild dementia in Hong Kong. *Int J Geriatr Psych* 2010;25(4):395-402. doi: 10.1002/gps.2352
4. Young DKW, Ng PYN, Kwok T, et al. The effects of an expanded cognitive stimulation therapy model on the improvement of cognitive ability of elderly with mild stage Dementia living in a community - a randomized waitlist controlled trial. *Aging Ment Health* 2019;23(7):855-62. doi: 10.1080/13607863.2018.1471586
5. Livingston G, Huntley J, Sommerlad A, et al. Dementia prevention, intervention, and care: 2020 report of the Lancet Commission. *Lancet* 2020;396(10248):413-46. doi: 10.1016/S0140-6736(20)30367-6
6. Baumgarten M, Battista RN, Infanterivard C, et al. The psychological and physical health of family members caring for an elderly person with dementia. *J Clin Epidemiol* 1992;45(1):61-70. doi: 10.1016/0895-4356(92)90189-T
7. Lau J, Ioannidis JP, Schmid CH. Quantitative synthesis in systematic reviews. *Ann Intern Med* 1997;127(9):820-6. doi: 10.7326/0003-4819-127-9-199711010-00008 [published Online First: 1998/02/12]
8. Reeves BC, Deeks JJ, Higgins J. 13 Including non-randomized studies. *Cochrane handbook for systematic reviews of interventions* 2008;1:391.
9. Salcher-Konrad M, Naci H, McDaid D, et al. Effectiveness of interventions for dementia in low- and middle-income countries: protocol for a systematic review, pairwise and network meta-analysis. *BMJ Open* 2019;9(6):e027851. doi: 10.1136/bmjopen-2018-027851
10. Popay J, Roberts H, Sowden A, et al. Guidance on the conduct of narrative synthesis in systematic reviews. A product from the ESRC Methods Programme 2006;1:b92.
11. Cuijpers P, Noma H, Karyotaki E, et al. Effectiveness and acceptability of cognitive behavior therapy delivery formats in adults with depression: a network meta-analysis. *JAMA Psychiatry* 2019;76(7):700-07. doi: 10.1001/jamapsychiatry.2019.0268
12. Chaimani A, Higgins JPT, Mavridis D, et al. Graphical Tools for Network Meta-Analysis in STATA. *Plos One* 2013;8(10) doi: 10.1371/journal.pone.0076654
13. Higgins JPT, Jackson D, Barrett JK, et al. Consistency and inconsistency in network meta-analysis: concepts and models for multi-arm studies. *Res Synth Methods* 2012;3(2):98-110. doi: 10.1002/jrsm.1044
14. Popay J, Roberts H, Sowden A, et al. Guidance on the conduct of narrative synthesis in systematic reviews. A product from the ESRC methods programme Version 2006;1:b92.
15. Cramer A, von Wyl A, Koemeda M, et al. Sensitivity analysis in multiple imputation in effectiveness studies of psychotherapy. *Front Psychol* 2015;6 doi: 10.3389/fpsyg.2015.01042
16. Sterne JAC, Savovic J, Page MJ, et al. RoB 2: a revised tool for assessing risk of bias in randomised trials. *BMJ* 2019;366:l4898. doi: 10.1136/bmj.l4898 [published Online First: 2019/08/30]

VERSION 2 – REVIEW

REVIEWER	Ma, Kris Pui Kwan University of Washington
REVIEW RETURNED	21-Jun-2021
GENERAL COMMENTS	The authors are to be commended in addressing many, if not most, of my earlier comments. The following are minor revisions that would be important to address before publishing:

	Minor revisions  • Population section – first paragraph, second line: spell out MCI because it is the first time to appear in the paper. • Population section, second paragraph: “We will also include studies conducted with people with diseases co-occurring with dementia and studies not specifying the type of dementia. However, we will exclude studies without explicating the diagnostic criteria for dementia or MCI when recruiting participants.” These two sentences are a bit confusing. Do you mean your review include studies of patients with diagnosed dementia of any types and other comorbidities, however exclude studies of patients who have cognitive impairments due to other medical conditions and do not have an explicit diagnosis of dementia? • Population section, third paragraph: “...lives with the person they care for, or provide direct or indirect care” – spelling error, should be “indirect” care • Comparison paragraph – you may want to repeat here saying this is a meta-analysis on RCTs, and therefore the following types of comparisons within the context of RCT studies are eligible for inclusion. • Data collection process section, second paragraph: “...which will be recorded in English based on the English abstract if available or personal translation” – a better word of choice might be “manual” translation. • Outcomes and prioritization section: “We will evaluate and compare the effectiveness of dementia-related interventions for outcomes of interest to ensure feasibility. However, the outcome of interest may need to change subject to the characteristics and the number of included studies.” These sentences are a bit confusing. Do you mean ‘Our study will prioritize evaluating and comparing the following outcomes of interests: XXX, XXX, XXX; However, we are also open to examine other outcomes that are being evaluated in the included studies’? Also, the use of ‘primary and secondary outcomes of interest’ in different parts of this section can be confusing, because they seem to mean different things in different contexts (i.e., primary and secondary outcomes of interest in your review versus primary and secondary outcomes of interest in original included study) • Subgroup analysis and meta-regression section, first paragraph and first line: “We will calculate the Cochran’s Q statistic and the I2 statistic to estimate the heterogeneity of the included studies” – spelling error, should be “Cochrane”. Additional suggestion  • Depending on stages of dementia among PLWD and assessment types, some assessments of PLWD’s cognition and functioning are self-reported by PLWD while others are reported by carers. It may be worthwhile to extract this data as well in the data extraction process and consider whether this should be included in analyses when evaluating the evidence across studies and interventions.
--	--

VERSION 2 – AUTHOR RESPONSE

Reviewer: 2

Dr. Kris Pui Kwan Ma, University of Washington

Comments to the Author:	Author responses
The authors are to be commended in addressing many, if not most, of my earlier comments. The following are minor revisions that would be important to address before publishing:	Thank you very much for your recognition and detailed comments. We have incorporated all of them in the revised version of the manuscript. Please kindly note that the changes in the revised version of the manuscript are highlighted using the track changes mode.
Minor revisions	
1. Population section – first paragraph, second line: spell out MCI because it is the first time to appear in the paper.	Thank you. As the term mild cognitive impairment (MCI) has been mentioned in the Introduction section (p6, para2), we used the acronym 'MCI' in the methods and analysis section.
2. Population section, second paragraph: “We will also include studies conducted with people with diseases co-occurring with dementia and studies not specifying the type of dementia. However, we will exclude studies without explicating the diagnostic criteria for dementia or MCI when recruiting participants.” These two sentences are a bit confusing. Do you mean your review include studies of patients with diagnosed dementia of any types and other comorbidities, however exclude studies of patients who have cognitive impairments due to other medical conditions and do not have an explicit diagnosis of dementia?	Thank you very much for pointing this out. To avoid confusion, we have revised the sentence “However, we will exclude studies without explicating the diagnostic criteria for dementia or MCI when recruiting participants” to “We will also include studies conducted among people with diseases co-occurring with dementia or MCI, and people with dementia or MCI with unknown subtype, as long as the diagnostic criteria for dementia or MCI were explicated.” (p8, para 3)
3. Population section, third paragraph: “...lives with the person they care for, or provide direct or indirect care” – spelling error, should be “indirect” care	Thank you. We have corrected the spelling error (p9, para1).
4. Comparison paragraph – you may want to repeat here saying this is a meta-analysis on RCTs, and therefore the following types of comparisons within the context of RCT studies are eligible for inclusion.	Thank you for this suggestion. To address this, we have completed the sentence as “Given the broad range for interventions of interest, any comparisons within the context of eligible study design will be acceptable for inclusion, such as active comparators, treatment as usual, placebo, and no treatment.” (p9, para3) We would like to specify RCT as the eligibility criteria

	later following the order of PICOS. We hope this is acceptable.
5. Data collection process section, second paragraph: "...which will be recorded in English based on the English abstract if available or personal translation" – a better word of choice might be "manual" translation.	Thank you for pointing it out. We have replaced it as 'manual translation' (p15, para2).
6. Outcomes and prioritization section: "We will evaluate and compare the effectiveness of dementia-related interventions for outcomes of interest to ensure feasibility. However, the outcome of interest may need to change subject to the characteristics and the number of included studies." These sentences are a bit confusing. Do you mean 'Our study will prioritize evaluating and comparing the following outcomes of interests: XXX, XXX, XXX; However, we are also open to examine other outcomes that are being evaluated in the included studies'? Also, the use of 'primary and secondary outcomes of interest' in different parts of this section can be confusing, because they seem to mean different things in different contexts (i.e., primary and secondary outcomes of interest in your review versus primary and secondary outcomes of interest in original included study)	Thank you and we apologise for the confusion in our writing. Following your suggestions, we have made the following revisions: First, we deleted the confusing sentence "We will evaluate and compare the effectiveness of dementia-related interventions for outcomes of interest to ensure feasibility. However, the outcome of interest may need to change subject to the characteristics and the number of included studies." and then added a new sentence in p16 para3: "Where feasible, we will also be open to examining other outcomes evaluated in the included studies." Second, we deleted all expressions of primary and secondary outcomes for our outcomes of interests to avoid confusion (p15, para 4; p16 para1&4).
7. Subgroup analysis and meta-regression section, first paragraph and first line: "We will calculate the Cochran's Q statistic and the I2 statistic to estimate the heterogeneity of the included studies" – spelling error, should be "Cochrane".	Thank you for pointing it out. We have now corrected this typo (p20, para1).
Additional suggestion	
8. Depending on stages of dementia among PLWD and assessment types, some assessments of PLWD's cognition and functioning are self-reported by PLWD while others are reported by carers. It may be worthwhile to extract this data as well in the data extraction process and consider whether this should be included in analyses when evaluating the evidence across studies and interventions.	Thank you very much for this helpful suggestion. This information will be extracted as details of the specific instrument. We will consider the potential effects of self-reported vs informant-reported vs standardised objective assessments in our analyses.